# Identification of Selection Signatures and Loci Associated with Important Economic Traits in Yunan Black and Huainan Pigs

**DOI:** 10.3390/genes14030655

**Published:** 2023-03-05

**Authors:** Yachun Han, Tao Tan, Zixin Li, Zheng Ma, Ganqiu Lan, Jing Liang, Kui Li, Lijing Bai

**Affiliations:** 1Shenzhen Branch, Guangdong Laboratory for Lingnan Modern Agriculture, Genome Analysis Laboratory of the Ministry of Agriculture, Agricultural Genomics Institute at Shenzhen, Chinese Academy of Agricultural Sciences, Shenzhen 518124, China; 2College of Animal Science & Technology, Guangxi University, Nanning 530003, China

**Keywords:** signatures of selection, pig high-density SNP chip, QTL, annotation

## Abstract

Henan Province is located in central China and rich in domestic pig populations; Huainan (HN) pigs are one of three Henan indigenous breeds with great performance, including early maturation, strong disease resistance and high meat quality. Yunan (YN) black pigs are a typical, newly cultivated breed, synthesized between HN pigs and American Duroc, and are subjected to selection for important traits, such as fast growth and excellent meat quality. However, the genomic differences, selection signatures and loci associated with important economic traits in YN black pigs and HN pigs are still not well understood. In this study, based on high-density SNP chip analysis of 159 samples covering commercial DLY (Duroc × Landrace × Large White) pigs, HN pigs and YN black pigs, we performed a comprehensive analysis of phylogenetic relationships and genetic diversity among the three breeds. Furthermore, we used composite likelihood ratio tests (CLR) and F-statistics (Fst) to identify specific signatures of selection associated with important economic traits and potential candidate genes. We found 147 selected regions (top 1%) harboring 90 genes based on genetic differentiation (Fst) in the YN-DLY group. In the HN-DLY group, 169 selected regions harbored 58 genes. In the YN-HN group, 179 selected regions harbored 77 genes. In addition, the QTLs database with the most overlapping regions was associated with triglyceride level, number of mummified pigs, hemoglobin and loin muscle depth for YN black pigs, litter size and intramuscular fat content for HN pigs, and humerus length, linolenic acid content and feed conversion ratio mainly in DLY pigs. Of note, overlapping 14 tissue-specific promoters’ annotation with the top Fst 1% selective regions systematically demonstrated the muscle-specific and hypothalamus-specific regulatory elements in YN black pigs. Taken together, these results contribute to an accurate knowledge of crossbreeding, thus benefitting the evaluation of production performance and improving the genome-assisted breeding of other important indigenous pig in the future.

## 1. Introduction

Pork is one of the most important forms of animal protein, with common use as a human dietary protein. Western commercial (Europe and America) pig breeds are well-known for superior performance, with fast growth, effective feed conversion efficiency and high carcass yield [1]. On the other hand, eastern (Asia) pigs, especially Chinese indigenous breeds, are characterized by early maturation, high reproduction rate, strong disease resistance, tolerance to harsh environments and high-quality meat. Therefore, the crossbreeding of western breeds and Chinese indigenous pigs has become the leading practice in China for developing new populations. Huainan (HN) pigs are one of northern China’s oldest native breeds, with all-black color, large body size, high fertility, strong adaptability and good meat quality. Yunan (YN) black is a typical, newly hybrid pig breed, synthesized between HN pig and American Duroc, selected for important traits such as growth and excellent meat quality [2]. In 2008, YN black pigs were registered as a new breed by China National Livestock and Poultry Genetic Resources Committee.

The process of domestication and modern breeding practice, with the subsequent breed formation and artificial directional selection, has left detectable selection signatures in numerous regions of the pig genome [3,4,5,6,7]. These regions can be analyzed by applying various genomic and population statistical methods. Li et al. characterized for under selection of distinct genomic regions involved in hypoxia, olfaction, energy metabolism and drug response in Tibetan wild boars based on whole-genome re-sequencing [8]. Ai et al. found a 14-Mb region with low recombination rate on the X chromosome playing a key role in regional adaptations to high and low latitude environments within China [9]. Moreover, Frantz et al. analyzed pig domestication using over 100 genome sequences and demonstrated that domestic pigs have strong signatures of selection at loci that have contributed to behavior and morphology [10]. Ma et al. identified signatures of artificial selection and loci related to intramuscular fat content and feed conversion ratio traits in Duroc Pigs [11]. In addition, a number of studies have been carried out to explore specific signatures of selection in Chinese indigenous pig breeds, including Anqing Six-End-White [12,13], Saba, Baoshan, Tunchang, Dingan [14], Meishan [15] and Lulai pigs [16]. Qiao et al. analyzed genetic structures and mapped genomic regions affecting backfat thickness of Yunan black pigs [17]. However, the genomic differences and integrated analysis of economical traits of HN and YN black pigs are still not well understood.

In this study, we analyzed 159 genotyped pigs, including one commercial breed (DLY), one Chinese indigenous breed (HN) and one Chinese cultivated breed (YN) with the 70 k Porcine Functional Variants Chip. SNPs were called and allele frequencies were calculated. We performed a comprehensive analysis of phylogenetic relationships and genetic diversity among the three breeds. Furthermore, two different statistics, the composite likelihood ratio (CLR) and fixation index (Fst) were used to identify specific signatures of selection associated with important economic traits and potential candidate genes in these three breeds. Our findings facilitate a better understanding of HN and YN black pigs’ genome characteristics and provide novel insights for molecular-assisted breeding strategies and germplasm conservation in the near future.

## 2. Materials and Methods

### 2.1. Ethics Approval

All animals used in this study were sampled according to guidelines for the Care and Use of Experimental Animals established by the Ministry of Agriculture and Rural Affairs of China. The protocol was approved by the College of Animal Science & Technology, Guangxi University (ethics approval reference number: Gxu-2022-114).

### 2.2. DNA Sample Collection

A total of 159 subjects from three pig populations were collected from China. Specifically, 99 commercial Duroc × Landrace × Large White (DLY) pigs, collected from Guangxi University, Nanning City, Guangxi Province, China. Additionally, 30 Huainan pigs (HN) and 30 Yunan black pigs (YN) were collected from Gushi Country, Xinyang City, Henan Province, China.

### 2.3. Single-Nucleotide Polymorphism (SNP) Genotyping and Data Quality Control

Genomic DNA samples from the three pig populations were extracted from ear tissue using the traditional standard phenol/chloroform method, and the quality was assessed using the light absorption ratio (A260/280). The Tianpeng chip (Porcine SNP70 Functional Variants Genotyping Array), which contains 187,244 SNPs, was used for the genotyping of all three pig populations. The SNP data can be downloaded from figshare database (https://figshare.com/articles/dataset/Pigs_70K_functional_locus_gene_chip/21900417, accessed on 14 January 2023). The sequence variants were called using PLINK v1.90 software [17,18], and quality control criteria were set as follows: (1) filtering out individuals with low detection rates according to the criterion of -mind 0.2; (2) removing SNP sites with a SNP deletion rate greater than 0.80 and unknown location or located on sex chromosome; and (3) removing SNP sites with minor allele frequency (MAF) less than 0.01. SNP genome coordinates were obtained from the Susscrofa 11.1 porcine genome reference assembly (https://www.ncbi.nlm.nih.gov, accessed on 12 September 2022).

### 2.4. Diversity, Phylogenetic, and Population Genetic Analyses

In order to gain a better understanding of the relationship among the three populations in this study, a phylogenetic tree (neighbor-joining tree, NJ-Tree) was constructed by using genetic distance matrix calculated by MEGA [19], then the tree was visualized online using the ITOL website (https://itol.embl.de accessed on 15 December 2022).

To investigate the pattern of genetic differentiation among different populations, principal component analysis (PCA) was performed on all autosomal SNPs using the smartpca program in the EIGENSOFT software package (https://mybiosoftware.com/eigensoft-population-structure-eigenanalysis-stratification.html accessed on 9 November 2022), significance of the eigenvalues was tested by Tracy–Widom distribution and the results were visualized using the R program ggplot2 package (https://www.R-project.org/ accessed on 28 November 2022).

For exploring the exactly genetic structure of the three populations, the population structure was deduced using the ADMIXTURE software v1.3.0 [20]. The K value from two to nine (K = 2–9) and the cross validation value (CV error) were calculated. To estimate and compare the linkage imbalance (LD) patterns of the three populations, PopLDdecay software (https://github.com/BGI-shenzhen/PopLDdecay accessed on 14 January 2023) was used to calculate the value of the squared correlation relationship coefficient (r^2^) between any two SNPs in a 300 kb interval [21].

### 2.5. Genome-Wide Selection Sweeps Detection

The genome-wide selection signatures of the three populations were identified using two different statistical methods: composite likelihood ratio tests (CLR) and F-statistics (Fst). The CLR values of the three populations were calculated separately using sweepfinder2 (v1.24) software [22]. The quality control criteria were set as follows: all CLR values of autosomes were rank ordered separately with a 10 kb sliding window and 10 kb step size, the top 1% was considered as regions under selection, and then these candidate regions were genetically, functionally annotated using ANNOVAR (v2.1.1) software [23].

Estimations of Fst among the three populations were calculated using the VCFTOOLS (v0.1.13) package [24] with a 200 kb sliding window and 20 kb step size, accompanied with minor 20 variants per window. All Fst values were sorted, the top 1% level was considered as putative regions under selection. The Fst figure was drawn using R (v4.3.1) package [25].

### 2.6. Annotation of Promoters of Tissue-Specific Expression Genes in 14 Tissues

In this study, we overlapped the top 1% regions sorted according to the Fst values of YN-DLY, YN-HN and HN-DLY groups with the 14 tissue-specific promoters’ annotation database [26].

### 2.7. Annotation of the Candidate Genes and Quantitative Trait Loci (QTL) Overlapping with Potential Selection Signatures

Genome annotation is based on Susscrofa 11.1 (https://www.animalgenome.org/blast/ accessed on 14 November 2022). Genes in these potentially selected regions were considered as candidate genes after RNAs and unidentified genes were filtered out. To explore the function of these identified genes, Gene Ontology (GO) [27] and Kyoto Encyclopedia of Genes and Genomes (KEGG) [28] were used for enrichment analyses through the Database for Annotation, Visualization, and Integrated Discovery (DAVID) Version 6.8 [29,30]. Furthermore, the pig quantitative trait loci (QTL) database [31] (https://www.animal-genome.org/cgi-bin/QTLdb/SS/index, on SS11.1 in gff format, accessed on 13 December 2022) was used to annotate economic traits associated with the selective sweep regions.

## 3. Results

### 3.1. Genotypes and Population Genetics Analysis

A total of 187,244 SNPs located on autosomes were genotyped in the SNP-chip. In the process of quality control, 65,779 SNPs and 2 individuals were discarded due to low SNP call rate and MAF. After the quality control, 121,465 SNPs from 157 individuals were retained for further study.

To describe the exact relationship among YN black, HN and DLY pigs, we first performed a neighbor-joining (NJ) tree analysis. The phylogenetic tree showed that the three populations were clearly divided into separate groups (Figure 1a). Secondly, PCA was conducted with PC1 and PC2, explaining 18.05% and 3.48% of the variations, respectively (Figure 1b). The PCA analysis clearly separated DLY from HN, with YN black dispersed in-between these two populations, as expected from the hybrid nature of YN black pigs. ADMIXTURE analyses were conducted with K = 2 to K = 9 (Appendix A) and cross validation analysis showed K = 8 was the optimal number of possible ancestral varieties (Figure 1c). When K = 2, all samples were divided into commercial pigs and Chinese indigenous pigs, both YN black and DLY pigs were clustered together. When K = 3, the YN black population could be well-distinguished from commercial pigs. When K = 8, the DLY pigs were separated into several sub-populations, and the YN population appeared a mixture of HN and DLY sub-populations (Figure 1d and Appendix A). Estimation of LD (r2) demonstrated highest mean LD and slowest decay in the DLY pigs and lowest mean LD and fastest decay in the YN black pigs (Figure 1e).

### 3.2. Identification of Signatures of Selection within-Population by CLR

To study the positive selection within the three populations, whole genome scans were performed by CLR and the top 1% was selected as the candidate regions. YN black pigs yield 2501 regions (Appendix A), HN pigs yield 2583 regions (Appendix A) and 2114 regions were found for DLY pigs (Appendix A). A total of 2829, 3242 and 3603 genes were harbored in these potential selection regions in YN, HN and DLY populations, respectively (Figure 2a–c).

GO terms enrichment were applied on the genes in the significant regions, and the top 20 *p*-value of each group are shown in Figure 2f. The most significant enriched terms of YN black pigs were insulin secretion (GO: 0030073, *p* = 0.00000426, 12 genes), cellular response to organic substance (GO: 0071310, *p* = 0.001716052, 273 genes) and negative regulation of peptide secretion (GO: 0002792, *p* = 0.000210512, 12 genes) (Figure 2d). The most significant enriched terms of HN pigs were regulation of primary metabolic process (GO: 0044238, *p* = 0.03215, 13 genes), cellular developmental process (GO: 0048869, *p* = 0.0002916, 608 genes) and regulation of metabolic process (GO: 0019222, *p* = 0.0000971, 1006 genes) (Figure 2e). For DLY pigs, the most significant enriched term was positive regulation of metabolic process (GO: metabolic process, *p* = 0.0000928, 10 genes) (Figure 2f).

### 3.3. Identification of Signatures of Selection between-Population by Fst

To further explore the specific signatures of selection between the three populations, Fst statistical analyses were conducted. We calculated Fst per site and averaged them in a non-overlapping 200 kb sliding window approach with a 20 kb step size across all autosomes. The windows where the scores of the statistics fell in the top 1% were considered significant. For the YN-DLY comparison, with DLY pigs treated as reference population, 147 of 14,804 sliding windows were identified as potential selection signatures, including 90 candidate genes (Figure 3a, Appendix A). For the HN-DLY comparison, with DLY pigs as the reference population, 169 out of 16,954 sliding windows were identified, including 58 genes (Figure 3b, Appendix A). Finally, for the YN-HN comparison, with HN pigs as the reference population, 179 out of 17,961 sliding windows were identified, including 77 genes (Figure 3c, Appendix A).

GO terms and KEGG pathway enrichment analyses on candidate genes were conducted. In the GO analysis, we found a total of 6 terms which were significantly enriched in YN black pigs (e.g., metabolic process (GO: 0008152, *p* = 0.093663, 17 genes), developmental process (GO: 0.032502, *p* = 0.36310, 11 genes) and growth (GO: 0040007, *p* = 0.039897, 2 genes)). The KEGG pathway analysis suggested that the candidate genes of YN black pigs were enriched in Tryptophan metabolism, cysteine methionine metabolism and inositol phosphate metabolism (Figure 4a). For HN pigs, there were 7 GO terms significantly enriched: metabolic process (GO: 0008152, *p* = 0.70383, 29 genes), developmental process (GO: 0032502, *p* = 0.41900, 15 genes), immune system process (GO: 0040007, *p* = 0.0245, 9 genes) and reproduction (GO: 0002376, *p* = 0.93621, 3 genes), among others. The KEGG pathway analysis showed that the HN pigs’ candidate genes were enriched in the JAK-STAT signaling pathway, vitamin B6 metabolism and fatty acid biosynthesis (Figure 4b). Finally, 5 GO terms were significantly enriched in DLY pigs, relating to metabolic process (GO: 0008152, *p* = 0.03215, 12 genes), growth (GO: 0040007, 0.069601, 2 genes), developmental process (GO: 0032502, 0.015029, 21 genes) and reproductive process (GO: 0022414, *p* = 0.068548, 3 genes). The KEGG pathway analysis indicated that the candidate genes of DLY pigs were enriched in alanine, aspartate and glutamate metabolism, phosphonate and phosphonate metabolism and biosynthesis of amino acids (Figure 4c).

To gain a better understanding of the mechanisms by which non-coding elements in selective regions cause quantitative trait differentiation between two different populations, we identified 14 tissue-specific promoters in the top 1% regions, sorted according to Fst value. Interestingly, we found much more muscle-specific regulatory elements in active promoters of YN black pigs’ selective regions compared to DLY pigs and HN pigs. In addition, the ratio of hypothalamus-specific regulatory elements in active promoters of the YN population’s selective regions was also the highest when compared to the other two (Figure 4d).

### 3.4. QTL Overlapping with Potential Signatures of Selection

We employed the pig quantitative trait loci (QTL) database to annotate economic traits associated with the top 1% Fst candidate regions. QTL overlapping with the candidate selection regions of YN black pigs was associated with triglyceride level, number of mummified pigs, hemoglobin and loin muscle depth. QTL overlapping with the candidate selection regions of HN pigs was associated with litter size and intramuscular fat content. For DLY pigs, the associated QTL traits were ear weight, humerus length, linolenic acid content and feed conversion ratio (Table 1).

## 4. Discussion

Henan is one of China’s central provinces, and includes one of the earliest pig domestication areas, with a rich diversity of indigenous pig breeds. As an excellent, newly cultivated black pig breed, Yunan (YN) black pigs are characterized by desirable body shape, tender local pig flavor and robust disease resistance. In the present study, we analyzed the population genetic diversity of YN black, HN and commercial DLY pigs. The results revealed that genetic stratification has occurred in these three populations. As expected from their hybrid nature, the YN black pig population displayed a genetic variation in-between DLY and HN pigs. The genetic diversity of a population is always recognized as a key factor in ensuring the survival and/or evolution of a species. Our results indicated Yunan black pigs have a richer diversity and are thus a valuable contribution to China’s germplasm bank.

In this study, we employed CLR and Fst to explore selection footprints within and between populations. The CLR method uses the combined likelihood of multiple markers to detect the selected sweeps in genomic regions [3]. CLR is especially robust in scenarios where changes in allele frequency occur fast [32]. The Fst method is classical and effective for detecting selection footprints in a single locus, based on population differentiation [33]. Voight et al. advised empirical cutoffs using the top 1% or 5% genome-wide, on all autosomes, to determine the significance of test statistic [34]. We adopted the same principle of data filtering to ensure high accuracy of the two methods in the present study, our results showed this strategy was more reasonable.

Selection signals can reflect loci and harboring genes that explain potential genetic mechanisms of phenotype polymorphism and provide scientific interpretation of the consequence of long-term domestication/breeding/evolution. In the results of CLR, we found that more genes related to fatty acid biosynthesis were selected in YN black pigs. We performed an extensive and accurate search of the human genome literature. One of the most interesting genes, LPIN1, acts as a magnesium-dependent phosphatidate phosphatase enzyme, which catalyzes the conversion of phosphatidic acid to diacylglycerol during triglyceride, phosphatidylcholine and phosphatidylethanolamine biosynthesis, and therefore controls the metabolism of fatty acids at different levels [35,36]. Additionally, the low-density lipoprotein receptor-related protein 5 (LRP5) gene may activate the canonical Wnt signaling pathway that controls cell fate determination and self-renewal during embryonic development and adult tissue regeneration [37]. For HN pigs, FOU1F1 is a pituitary-specific transcription factor which plays an essential role in the specification of the somatotroph, lactotroph and thyrotroph lineages, and specifically activates GH1, PRL and TSHβ transcription [38]. FBOX31 has a role in regulating the cell cycle as well as dendrite growth and neuronal migration [39]. Interestingly, genes related to growth were selected by a more intense artificial selection in the DLY pig breed; CCN3,essential for human hematopoietic stem or progenitor cells functional integrity [40], plays an important role in cardiovascular, skeletal development and fibrosis [41,42]. MORC is testis-specific in mouse but transcribed in multiple somatic tissues in human, and serves important biological functions in both meiotic and mitotic cells [43]. Meanwhile, the result of Fst has supported our findings mentioned above. As a newly cultivated pig breed of economic importance, leanness and flavor has consistently been considered as an objective trait of YN black pigs. When comparing YN black with DLY pigs, several genes were found related to growth rate: SP6 gene caused severe hypoplastic amelogenesis imperfecta [44] and LCORL was reported to be associated with adult height in individuals of African ancestry [45]. When comparing YN black with HN pigs, several selected genes were found to be related to brain and neuron development. PDZD8 has been shown to be required for calcium ion (Ca^2+^) uptake in mammalian neurons [46]. Maternal embryonic leucine zipper kinase (MELK) regulates multipotent neural progenitor proliferation [47]. When comparing HN with DLY pigs, more selected genes were found to be related to metabolic procession. ACSF3 deficiency was the first human disorder identified and caused human metabolic disorders [48]. Furthermore, ACSF3 protein is a malonyl-CoA that could be used for de novo fatty acid synthesis in mammalian mitochondria [49]. PIEZO1 plays a critical role via intracellular accumulation of glucose metabolites and regulating insulin secretion [50].

The functional enrichment analysis of genes usually only focuses on biological information, hence the intrinsic relationship between genes and important quantitative traits is usually less conclusive. QTL mapping can serve as a reference or potential clue to elucidate the identified signatures of selection. In this study, the pig QTL database and enrichment analyses were used to annotate economic traits associated with overlapped selective regions. Through the overlapped results, we confirmed the detected selection regions were not only related to fat deposition, but also to growth, and meat quality and fertility in YN black pigs and HN pigs, respectively.

## 5. Conclusions

In summary, we used the 70 k Porcine Functional Variants Chip to detect the heritable variation information among Yunan (YN) black pigs, Huainan (HN) pigs and DLY pigs. The characterization of genomic diversity and population structure demonstrated the direction for genetic assessment and the development of reasonable breeding strategies for YN black pigs. Moreover, CLR analysis was implemented for selective sweep detection within-populations and Fst estimation was conducted for selective signatures detection between-populations. Both methods identified several selective regions in accompanied representative groups associated with important quality traits. In addition, overlapping 14 tissue-specific promoters’ annotation with the Fst top 1% selective regions systematically described the muscle-specific and hypothalamus-specific regulatory elements in YN black pigs specifically. Taken together, these results provide a valuable resource for further genetic research to improve the genome-assisted breeding of other important indigenous pigs in the future.

## Figures and Tables

**Figure 1 genes-14-00655-f001:**
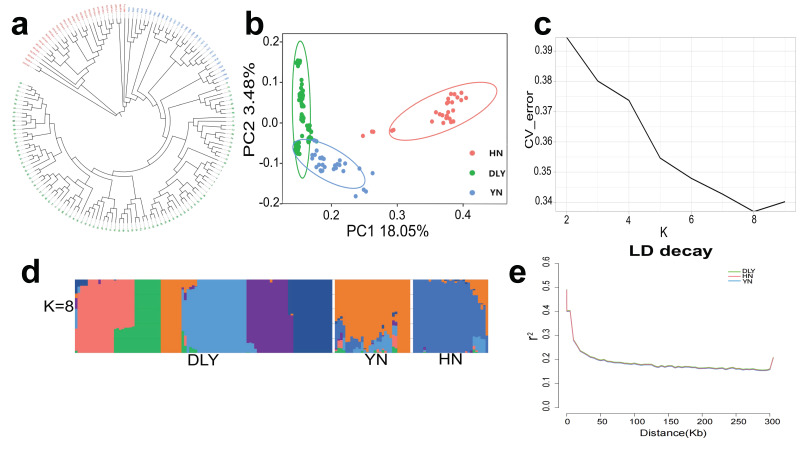
Genomic characteristics of three pig breeds. (**a**) Neighbor-joining (NJ) tree of the after-filtering 157 pigs covering 30 YN black pigs, 28 HN pigs and 99 DLY pigs. Green color: DLY pigs; blue color: YN black pigs; red color: 28 HN pigs. (**b**) Principal component analysis of 30 YN black pigs (blue), 28 HN pigs (red) and 99 DLY pigs (green). (**c**) Ancestral composition analysis K = 2–9. (**d**) Structure analysis with K = 8. (**e**) LD decay in different breeds with 300 kb.

**Figure 2 genes-14-00655-f002:**
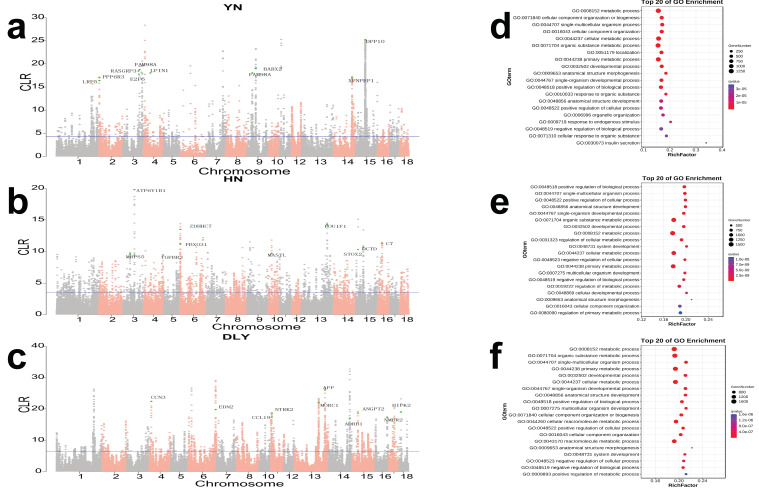
Genome-wide distribution of selection signatures detected by CLR. (**a**) CLR statistics across all autosomes in YN black pigs. (**b**) CLR statistics across all autosomes in HN pigs. (**c**) CLR statistics across all autosomes in DLY pigs. The blue line corresponds to the 99% threshold on the corresponding empirical distributions. (**d**–**f**) The top 20 GO enrichment analysis of selected genes in YN black pigs, HN pigs and DLY pigs.

**Figure 3 genes-14-00655-f003:**
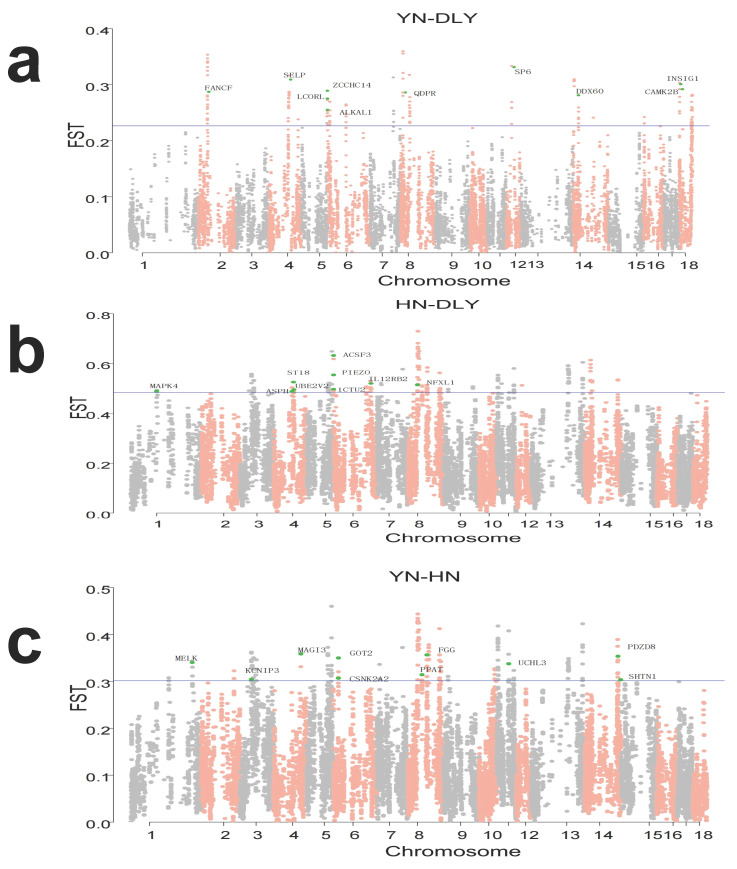
Genome-wide distribution of selection signatures detected by Fst. (**a**) Fst statistics across all autosomes in YN black pigs. (**b**) Fst statistics across all autosomes in HN pigs. (**c**) Fst statistics across all autosomes in DLY pigs. The blue line corresponds to the 99% threshold on the corresponding empirical distributions.

**Figure 4 genes-14-00655-f004:**
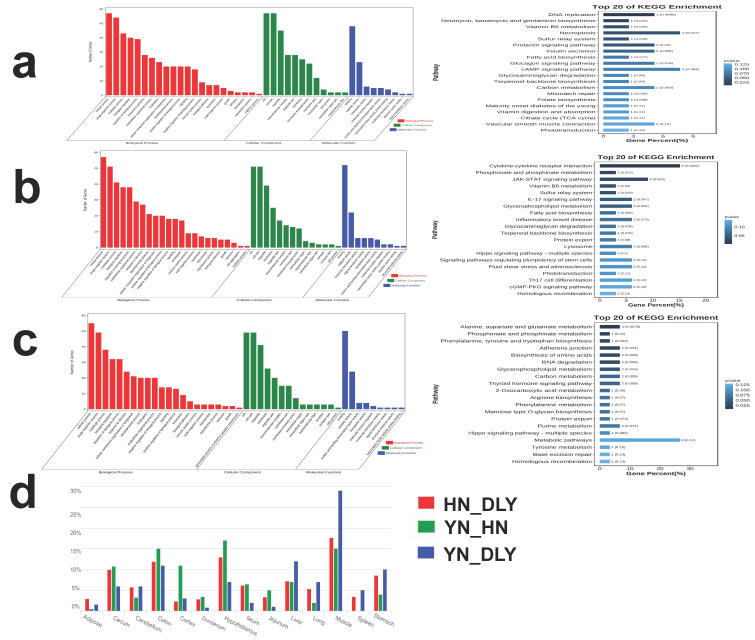
The function analysis of the selected genes. (**a**) KEGG pathway and top 20 GO enrichment analysis of differential expressed genes in YN-DLY group. (**b**) KEGG pathway and top 20 GO enrichment analysis of differential expressed genes in HN-DLY group. (**c**) KEGG pathway and top 20 GO enrichment analysis of differential expressed genes in YN-HN group. (**d**) Annotation of the non-coding selection regions with 14 tissues promoter database.

**Table 1 genes-14-00655-t001:** Genomic regions identified as signatures of selection in three breeds overlapped with QTL database.

Fst Top 1% Overlaps Candidate Regional Genes and Traits with Porcine QTL						
Group	CHR	START	END	FST	QTL	Genes
YN_DLY	Chr.1	26,823,251	26,823,255	0.93663199	Triglyceride level QTL (56,029)	DOLK, CRAT, PTPA, DOLPP1
	Chr.12	28,689,860	28,689,864	0.73554397	Hemoglobin QTL (22,161)	PNPO
	Chr.12	29,033,566	29,033,570	0.61874989	Loin muscle depth QTL (255,351)	NPEPPS, OSBPL7, SRCN2
	Chr.14	136,511,430	136,511,434	0.93663199	Number of mummified pigs QTL (178,885)	MYOF, CEP55, ADAM12,
YN_HN	Chr.8	38,946,696	38,946,700	0.93663199	Litter size QTL (130,289)	DCUN1D4, SPATA18, PAICS, PPAT
	Chr.11	47,893,515	47,893,519	0.61874989	Intramuscular fat content QTL (147,429)	UCHL3, KATNAL1
HN_DLY	Chr.1	100,049,958	141,109,887	0.93663199	Ear weight QTL (8853)	MAPK4
	Chr.4	77,799,657	77,799,661	0.93663199	Humerus length QTL (57,328)	UBE2V2
	Chr.4	79,493,441	79,493,445	0.61874989	Feed conversion ratio QTL (139,907)	ST18
	Chr.6	145,288,072	145,288,076	0.73554397	Linolenic acid content QTL (193,502)	PIEZO1, ACSF3, CTU2

## Data Availability

The SNP data can be downloaded from figshare database (https://figshare.com/articles/dataset/Pigs_70K_functional_locus_gene_chip/21900417, accessed on 14 January 2023).

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
