# Peer review of "Identification of Selection Signatures and Loci Associated with Important Economic Traits in Yunan Black and Huainan Pigs"

_genes, 2023, doi:10.3390/genes14030655_

Round 1

Reviewer 1 Report

line 17: Remove the first and in the list and add commas

Line 20: Define DLY

Are CLR, Fst, and QTL on the acronym list for this journal? Of not define

Line 28: Start new sentence when you start talking about HN pigs

Line 29: Also start a new sentence for DLY pigs

Line 33: re-word to make flow

Line 39-40: Reword- either say most important or main, this sentence contains redundancy.

Line 44: strong disease resistance, tolerance to harsh environments (remove the and)

Line 49: replace tasty with a more scientific word

Line 54: make practice plural 

Line 57: remove of

Line 58: Remove and approaches

Line 58-60: What did Li et al find??

Line 65: Change has to have

Lines 75-85: This belongs in the materials and methods. In its place please put an objectives statement. 

Lines 105-106: need some punctuation and spacing around the website

Lines 251-258: Paragraph duplicated remove one of them

277: need a comma after nature

Line 278: DLY rather than duroc

Discussion: Please bridge the gaps between the literature and your data. I assume you are trying to tie in some of that human genome work to relate to the pigs genes, but I am not seeing the bridge as I read it. 

I think the conclusion is good, but there is too much analytical description in the lines 338 and 339. I suggest you remove that 

***Please fix all grammar throughout, I pointed out some but there is plenty more***

Reviewer 2 Report

Overall, the manuscript is interesting. I propose to slightly change the introduction and the first sentences of the abstract. Please note that this paper is not about Chinese pig breeds, but about identification of selection signatures.

1. There is no justification for the choice of experimental material. What are the reasons for such large disproportions in the size of the three groups?

2. I have the impression that the authors are confusing the terms "genome" and "genotype" (lines: 75-76). The genome relates to species, not individuals. So I suggest you write" "...we analyzed 159 genotyped pigs..."

3. Please remember that a computer program is not a method!

4. There are some errors in citing other authors' publications.
